# A Deep-Learning Approach for Diagnosis of Metastatic Breast Cancer in Bones from Whole-Body Scans

**Nikolaos Papandrianos [1,\*]**, **Elpiniki Papageorgiou [2]**, **Athanasios Anagnostis [3,4]** and **Anna Feleki [5]**

[1]   Former Nursing Department, University of Thessaly, 35100 Lamia, Greece
[2]   Faculty of Technology, University of Thessaly, Geopolis Campus, Larissa—Trikala Ring Road, 41500 Larissa, Greece; elpinikipapageorgiou@uth.gr
[3]   Institute for Bio-economy and Agri-technology (iBO), Center for Research and Technology—Hellas (CERTH), 57001 Thessaloniki, Greece; a.anagnostis@certh.gr
[4]   Department of Computer Science, University of Thessaly, 35131 Lamia, Greece
[5]   Faculty of Technology, Geopolis Campus, University of Thessaly, 41500 Larissa, Greece; felekiann@gmail.com
\*   Correspondence: npapandrianos@uth.gr

**Abstract:** (1) Background: Bone metastasis is one of the most frequent diseases in breast, lung and prostate cancer; bone scintigraphy is the primary imaging method of screening that offers the highest sensitivity (95%) regarding metastases. To address the considerable problem of bone metastasis diagnosis, focused on breast cancer patients, artificial intelligence methods devoted to deep-learning algorithms for medical image analysis are investigated in this research work; (2) Methods: Deep learning is a powerful algorithm for automatic classification and diagnosis of medical images whereas its implementation is achieved by the use of convolutional neural networks (CNNs). The purpose of this study is to build a robust CNN model that will be able to classify images of whole-body scans in patients suffering from breast cancer, depending on whether or not they are infected by metastasis of breast cancer; (3) Results: A robust CNN architecture is selected based on CNN exploration performance for bone metastasis diagnosis using whole-body scan images, achieving a high classification accuracy of 92.50%. The best-performing CNN method is compared with other popular and well-known CNN architectures for medical imaging like ResNet50, VGG16, MobileNet, and DenseNet, reported in the literature, providing superior classification accuracy; and (4) Conclusions: Prediction results show the efficacy of the proposed deep learning approach in bone metastasis diagnosis for breast cancer patients in nuclear medicine.

**Keywords:** bone metastasis; breast cancer; whole body; scintigraphy; deep learning; image classification; convolutional neural networks

## 1. Introduction

Bone metastasis is cancer that initially started in another part of the body and spread to the bone. Various names can be found in the literature, such as secondary bone cancer or metastatic bone disease [1,2]. Bone metastasis commonly occurs in the spine, thigh and pelvis but far from that, it can also be seen in any bone. Sometimes, bone metastasis appears years after cancer treatment, or may be the first indication of cancer.

Metastatic cancer in bones is a life-threatening disease. Proper diagnosis and treatment of metastatic cancer in bones can save people's lives [3]. Identification of normal, benign, and malignant tissues seems to be a step of high importance before treatment of bone metastasis tumors. The most



common tumors, which frequently metastasize to bone, are those of the breast, lung and prostate, whereas the skeleton area especially in patients with advanced disease, accumulates most of the tumor burden [4]. In particular, breast cancer and prostate cancer are the most common types of cancer likely to spread to bone [3]. Previous studies clearly report that 60%–75% of metastatic breast cancers are initially diagnosed with bone metastasis [5,6].

Bone metastasis in women who suffer from breast cancer is clinically important [7], and has a significant impact on the quality of their life [8]. Rapid diagnosis of bone metastases can be achieved using modern imaging techniques such as scintigraphy, positron emission tomography (PET) and whole-body magnetic resonance imaging (MRI) [9–11]. Among them, bone scintigraphy offers the highest sensitivity (95%) regarding metastases, making it the primary imaging method of screening, [12–14]. In particular, whole-body bone scintigraphy emerges as the most common diagnostic procedure in nuclear medicine for clinical routine investigation [14–16] and can reveal the sites that bone metastases are distributed in all types of cancer, possibly predicting the outcome [17,18]. It actually presents high sensitivity whereas the changes of bone metabolism are identified by doctors and physicians earlier than the changes in bone structure detected by skeletal radiograms [2,7]. Infections, degenerative changes, benign or malignant diseases and other clinical entities are among indications in bone scintigraphy [16]. This method, however, shows low specificity, as it cannot distinguish between the causes of bone turnovers and those of metastatic origin (leukaemia, healing fracture, etc.), which is a significant drawback of this method.

At present, bone scintigraphy which uses single-photon emission computed tomography (SPECT) imaging, remains the most used and well-known imaging method for clinical diagnosis of metastatic cancer, both men and women [12,13,18]. It turned out to be an effective tool for early diagnosis of bone metastasis in breast and prostate cancer patients [15,17]. In particular, with the help of nuclear medical imaging of the targeted area of the body, which is used for the identification of benign and malignant conditions, the doctor and the physician are able to diagnose bone metastasis [10,19].

To address the considerable problem of bone metastasis diagnosis, machine learning (ML) methods focused on deep-learning algorithms for medical image analysis, have been recently investigated by many researchers. The advancements in the field of machine learning have led to more intelligent and robust computer-aided diagnosis (CAD) systems, as the learning ability of ML methods have been constantly improving [20,21]. Also, the groundbreaking performance of ML in CAD systems is based on deep learning (DL). DL has outstanding capabilities which make a very significant impact on improving the diagnostics potential of CAD systems in medical imaging [22–24].

A recent survey reveals the penetration of deep-learning techniques entirely into the field of medical image analysis in CAD; classification, detection, segmentation, registration, retrieval, image generation and enhancement, whereas the successful application of deep learning to medical imaging tasks is thoroughly examined [22–26]. Specifically, deep learning in medical imaging was achieved using convolutional neural networks (CNNs) [27–30]. CNNs appertain to deep, feed-forward artificial neural networks, able to extract the feature of an image and use this feature to classify the image. The aim of CNNs in all application fields is to convert the input into a feature vector by means of a matrix and to match it with trained feature vectors [31].

Some major and popular CNN models, introduced in published research articles in the context of image processing and deep learning, are: the AlexNet (2012) network [32], which has a very similar architecture to LeNet, by Yann LeCun et al., but is deeper, with more filters per layer and includes stacked convolutional layers, the VGGNet16 (2014), that consists of 16 convolutional layers and its very uniform architecture makes it very appealing [33], the ResNet model, that introduces the residual learning building block for extremely deep convolutional networks [34], the DenseNet (2017) [35] that offers the main advantage of alleviating the gradient vanishment problem with the direct connection of all the layers and, finally, the Deeplab, which deals with the introduction of convolutional layers atrous for semantic image segmentation in deep convolutional neural networks [36]. All the important

and most interesting applications of deep learning in medical image analysis and segmentation are gathered in recent review studies [22–24,29].

### 1.1. Related Research in Breast Cancer Diagnosis Using Convolutional Neural Networks (CNNs)

Regarding the medical image domain which deals with breast cancer diagnosis, previous studies examined the inclusion of machine learning approaches into breast cancer computer-aided diagnosis and image classification [29,37–41]. The state of the art on machine learning techniques for breast cancer computer aided diagnosis offers a wide range of analysis regarding the current status of CAD systems, when image modalities used and machine learning-based classifiers are taken into consideration [38,39,42,43]. Recent review studies in [44–47] presented a thorough survey on both traditional ML and DL literature with particular application in breast cancer diagnosis. Also, the bibliographic review in study [47] provided insightful characteristics of some well-known DL networks in breast cancer diagnosis, both in primary and metastatic breast cancer.

Investigating the area of computer-aided biomedical images applied in breast cancer diagnosis, there are some recent research studies which show the advantageous features of deep convolutional nets [48–50]. In particular, in [51] a CNN was applied for breast cancer histopathology images classification, in two different classes, benign and malignant. Suzuki et al. (2016) implemented CNNs for mass detection in mammographic images in computer-aided diagnosis [52]. Spanhol et al. (2016) applied CNNs for the classification of breast cancer from histopathological images [53], whereas Wichakam and Vateekul, (2016) combined deep convolutional networks and SVMs for mass detection on digital mammograms [54]. Swiderski et al. investigated the application of deep learning and non-negative matrix factorization for breast cancer recognition in mammograms [55]. In the same year, Kallenberg and his colleagues explored the capabilities of unsupervised deep learning to breast density segmentation and mammographic risk scoring [56]. Dealing with breast cancer image detection, CAD systems are fundamentally used in screening mammography for detection of mass [41,48,57,58]. In addition, CNNs have mostly been applied in mammography images and computed tomography (CT) images for classification and segmentation case studies [38,39,46–49]. Furthermore, deep CNNs and transfer-learning models have been applied for the prediction of lymph node metastasis in patients with primary breast cancer on the basis of US images [59], as well as on the basis of histopathological images [60].

It is evident that much research has already been done for the detection and diagnosis of primary breast cancer in the past few years. Some representative CNN applications for breast cancer diagnosis are summarized in Table 1.

**Table 1.** Summary of representative convolutional neural networks (CNNs) for breast cancer (BC) diagnosis.

| Type of BC | Application Type | Modality | Reference | Data Available |
|---|---|---|---|---|
| Primary | classification | Histopathological images | Kumar K. et al. [51] | NO |
| Primary | detection | Mammograms | Swiderski, B. et al. [55] | YES |
| Primary | classification | Histopathological | Spanhol, F.A. et al. [53] | NO |
| Primary | mass detection | Mammograms | Wichakam, I. et al. [54] | NO |
| Primary | mass detection | Mammograms | Suzuki, S. et al. [52] | NO |
| Primary | segmentation | Mammograms | Kallenberg, M. et al. [56] | NO |
| Primary | mass detection | Mammograms | Fenton J.J. et al. [58] | NO |
| Primary | mass detection and classification | Mammograms | Cheng H. et al. [41] | NO |
| Primary | classification | Mammograms | Chougrad, H et al. [57] | NO |
| Metastatic in lymph nodes | prediction | Ultrasound images | Zhou Li-Q. et al. [59] | NO |
| Metastatic in lymph nodes | detection and classification | Histopathological | Steiner D.F. et al. [60] | NO |

However, our work is devoted to the investigation of deep-learning algorithms on bone metastasis diagnosis in breast cancer patients using bone scintigraphy. In nuclear medicine, bone scintigraphy and PET/CT are the two most used methods for bone metastasis detection in breast cancer [61–64]. Concerning the involvement and applicability of CNNs in diagnosis of metastatic breast cancer in bones from whole-body scan images, no previous works in CNNs exploration are reported. There are only a few research studies applying neural networks to deal with breast cancer bone metastasis classification in nuclear medicine. In what follows, the related papers of metastatic breast cancer diagnosis in bones using neural network methods are discussed.

*1.2. Literature Review in Bone Metastasis Diagnosis from Breast Cancer Patients in Nuclear Medicine*

Focusing on bone scintigraphy, which is considered as the well-known imaging method of screening, concerning the diagnosis of bone metastasis, only two previous studies were reported in the literature that deal with metastatic breast cancer classification. The first study was devoted to the development of a computer-aided diagnosis system for bone scintigraphy scans (CADBOSS) [65]. CADBOSS is able to detect metastases in an accurate way, that combines an active contour segmentation algorithm for hotspots detection, an advanced method of image gridding to extract certain characteristics of metastatic regions and finally, an artificial neural network classifier for identifying possible metastases [65]. Another similar research study was conducted during a PhD thesis in [66] which deals with the investigation of CAD system to improve the diagnostic accuracy of planar whole-body bone scan interpretations. During this PhD thesis, four sub studies were accomplished sequentially. The first sub study investigated an automated method that determines the existence of metastases after bone scanning, using neural networks and image-processing techniques. The second study dealt with the discrepancies between observers as well as the performance, in terms of interpretations solely on bone scan images, regarding the existence of bone metastatic disease. The third study resulted in the development of a second CAD system based on improved image processing and artificial neural network to classify bone scans in two categories, considering a larger database of whole-body bone scans. Further investigation took place on how the proposed CAD system can benefit the physicians by reducing inter-observer variation. The final outcome of that thesis was the development of a totally automated computer-assisted diagnosis system that can identify metastases after examining bone scans, applying multi-payer perceptron artificial neural network techniques, and a small database of whole-body bone scans (135 patients). The highest sensitivity achieved from all the studies and accomplished during this thesis was approximately 89% [66]. None of these research studies provide publicly available data.

As there are no other research studies for breast cancer bone metastasis classification in nuclear medicine (and no research paper has been published, either in journals or conferences, which concerns bone scintigraphy analysis and classification, using deep learning), we briefly report some recent studies in nuclear medicine for bone metastasis diagnosis in men suffering from prostate cancer. Bone scintigraphy is a well-recognized image modality for metastatic prostate cancer diagnosis in bones.

*1.3. Literature Review in Bone Metastasis Classification Using CNNs in Nuclear Medicine*

As far as prostate cancer bone metastasis classification is concerned, most of the studies applying CNNs are devoted to PET/CT imaging and SPECT scans [67–76]. These studies have been gathered in Table 2.

**Table 2.** Summary of CNNs applied in whole-body scans for bone metastasis classification in nuclear medicine imaging.

| Application Type | Modality | Reference | Data Available |
|---|---|---|---|
| Metastatic prostate cancer classification | SPECT | [70] | NO |
| Metastatic prostate cancer classification | NaF PET/CT images | [71] | NO |
| Malignancy detection | FDG PET-CT | [72] | NO |
| Metastatic prostate cancer classification | FDG PET | [73] | NO |
| Patient's sex prediction | FDG PET-CT | [74] | NO |
| predicts whether physician's further diagnosis is required or not | FDG PET-CT | [75] | NO |
| Segmentation | SPECT | [76] | NO |

In [70], a master thesis, was the first study devoted to CNN application by correctly classifying metastasis in bone scintigraphy images for prostate cancer. That thesis focused mainly on classification problems without considering any identification and segmentation tasks. The used dataset was provided by Exini Diagnostics AB, in the form of image patches of hotspots already found. Due to time frame restrictions, only the hotspots found in the spine had been used to train the CNN, as those were considered the easiest to classify. Bone scan index (BSI) was calculated for whole-body bone scans, by segmenting the entire skeleton from the background in both the anterior and posterior views. A shape model based on a mean shape of several normal whole-body scans was fitted to the skeleton, using an image analysis algorithm called Morphon registration. The outcomes of this master thesis [70] showed that the calculated accuracy of the validation set was 0.875, whereas the calculated accuracy of the testing set was 0.89.

In PET/CT imaging, a deep convolutional neural network has been implemented to distinguish benign from malignant lesions in NaF PET/CT images of patients with metastatic prostate cancer [71]. Also, a CNN-based system was examined to detect malignant findings in FDG (fluorodeoxyglucose)PET-CT examinations in a retrospective study [72]. Various CNN models with equivalent configurations, such as ResNet, DenseNet and Visual Geometry Group (VGG), were used to classify whole-body FDG PET images in three categories of benign, malignant and equivocal patient cases. Location of malignant uptake was classified at the same time into, (A) head-and-neck, (B) chest, (C) abdomen, (D) pelvic region. Different predictions were derived for each location [73].

In another research work [74], a simple CNN model with an equivalent configuration to VGG, that predicts patient sex from FDG PET-CT images was proposed. Kawauchi and his colleagues in [75] proposed an alternative CNN-based diagnosis system for whole-body FDG PET-CT that predicts whether physician's further diagnosis is required or not. A thorough analysis of the results showed an accuracy of 93.2% ± 3.9% regarding images of patients with malignant uptake while the respective accuracy for images of equivocal were 87.8% ± 5.3%. Furthermore, the task of segmentation in skeletal scintigraphy images with deep learning models has been discussed in the study of [76], in which the authors examined different approaches to convert convolutional neural networks, designed for classification tasks, into powerful pixel-wise predictors. However, this research domain of bone scintigraphy segmentation has not been established well yet with the inclusion of advanced deep neural networks.

Reviewing the relevant literature for diagnosis of bone metastasis using whole body images, the authors notice that all previous works focus mainly on PET imaging and prostate metastatic bone cancer, while no research paper seems to have been published, dealing with bone scintigraphy analysis and classification, in patients with breast cancer, using CNNs.

### 1.4. Motivation and Aim of This Research Study

Although bone scintigraphy is extremely important for diagnosis of metastatic cancer in bones, in both men and women, there is currently no research paper regarding the classification of breast cancer metastasis in bones from whole-body images (bone scintigraphy) that uses efficient and robust CNNs.

Nowadays, the main challenge in bone scintigraphy, which is considered one of the most sensitive methods for imaging in nuclear medicine, regards building an algorithm that automatically identifies whether a patient is suffering from bone metastasis or not, only by looking at the whole-body scans. The algorithm has to be extremely accurate since people's lives could be at stake. Machine-learning approaches, with a focus on deep learning algorithms, have particularly shown a promising applicability in medical image analysis in the area of nuclear medicine. However, classification accuracy of bone scintigraphy, analyzed by machine-learning approaches, involving CNN systems, has not been well established yet.

The current work introduces an efficient CNN method for the diagnosis of bone metastasis disease, by identifying whether a patient is suffering from bone metastasis or not, based on whole-body images. After a thorough CNN exploration, taking into consideration both network architecture and hyperparameter configuration, the researchers propose a robust CNN to classify images in two classes, benign and malignant. A retrospective study involving 408 women patients suffering from breast cancer, was conducted to examine the accuracy of bone scintigraphy analyzed by CNNs. For validation purposes of the proposed classification methodology, a meticulous comparison analysis was performed between the proposed CNN method and a number of well-known and popular CNN architectures. In this work, popular and advanced CNN structures such as ResNet50, VGG16, and DenseNet that were previously proposed for medical image classification [22–24,30], are applied in the dataset, after their necessary parameterization and configuration. The experimental results reveal that the proposed method achieves superior performance as far as bone scintigraphy is concerned, outperforming other well-known methods of CNNs.

The novel contribution of this research work lies in the creation of a robust CNN model (being the most efficient deep learning method in medical image analysis) for bone metastasis diagnosis from breast cancer patients using whole-body scans. Following the literature review and the state-of-the-art deep learning methodology (as presented in Sections 1.1 and 1.2), it is obvious that there is no previous work on metastatic breast cancer identification in bones from whole-body scans using deep learning and CNNs.

The innovations and contributions of this paper are summarized as follows:

- The creation and demonstration of a customized, robust CNN-based classification tool for identification of metastatic breast cancer in bones from whole-body scans.
- The meticulous exploration of CNN hyper-parameter selection to define the best architecture and select the RGB mode selection that could lead to enhanced classification performance.
- The comparative experimental analysis performed for utilizing popular image classification CNN architectures, like ResNet50, VGG16, GoogleNET, etc.

The proposed methodology apparently improves the diagnostic effect of the deep-learning method. Moreover, after a thorough assessment of the results produced, the proposed approach is proven to be effective for classification of whole-body images in nuclear medicine.

This paper is structured in the following fashion: Section 2 presents the material and methods related to this research study. The proposed network based on CNN for bone metastasis diagnosis in breast cancer using whole body images is described in Section 3. The exploration analysis of CNNs in both RGB and grayscale modes is illustrated in Section 4, considering different parameters and configurations, thus providing the best CNN model for this case study. Section 4 also gathers all the performed experiments with the corresponding results, whereas Section 5 provides a thorough discussion about the analysis of results. The main outcomes of the paper are concluded in Section 6, which outlines the future steps, as well.

## 2. Materials and Methods

### 2.1. Breast Cancer Patient Images

This study was approved by the Director of the Diagnostic Medical Center "Diagnostico Iatriki A.E." and the requirement to obtain informed consent was also waived by the Director of the Diagnostic Center, due to the study's retrospective nature. All procedures in this work were in accordance with the Declaration of Helsinki. A retrospective review of 422 consecutive, previously selected whole-body scintigraphy images from 382 different patients (women) who visited the Nuclear Medicine Department of Diagnostic Medical Center "Diagnostico Iatriki A.E." in Larissa, Greece, between June 2013 and December 2017, was performed. The selection criterion were breast cancer patients who had undergone whole-body scintigraphy, because of suspected bone metastatic disease.

Due to the fact that there are various remains and non-osseous uptake presented in images (i.e., urine contamination and medical accessories), as well as the frequent visible site of radiopharmaceutical injection [77], a preprocessing approach was accomplished to remove these artifacts from the original images. This preprocessing method was accomplished by a nuclear medicine physician, before the use of the dataset in the proposed classification approach.

Next, the selected images were diagnosed accordingly by a Nuclear Medicine Specialist with 15 years' experience in bone scan interpretation. In total, 408 bone scan images were selected by the nuclear medicine specialist and classified accordingly into two categories (1) malignant and (2) benign; 221 bone scans concern malignant images whereas 187 bone scans concern benign images without bone metastasis (see Figure 1).

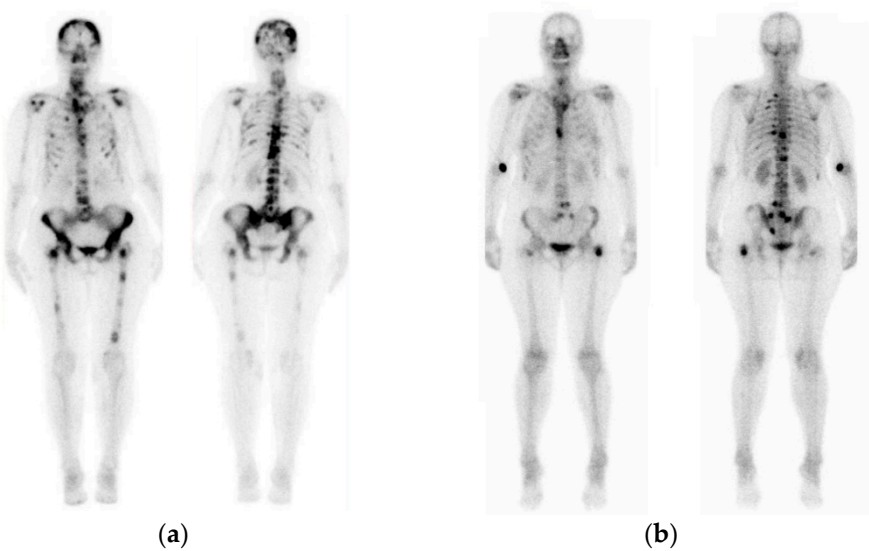

(**a**)　　　　　　　　　　　　　　(**b**)

**Figure 1.** *Cont.*

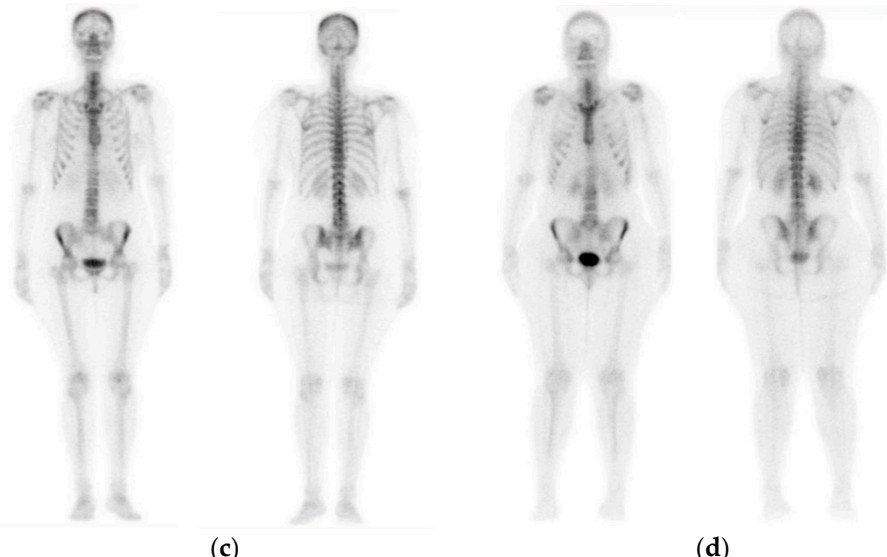

**Figure 1.** Image breast cancer bone metastasis samples in the dataset (Label: (**a,b**) malignant and (**c,d**) benign).

The equipment used for scanning patients is a Siemens gamma camera Symbia S series SPECT System (by dedicated workstation and software Syngo VE32B) with two heads with low-energy high-resolution (LEHR) collimators, where the speed of scanning is 12 cm/min with no pixel zooming. Regarding the procedure followed for bone scintigraphy, two types of radionuclide were used, the 99m-Tc-HDP (TechneScan®) and the 99-Tc-MDP (PoltechMDP 5mg). Approximately 3 h after an intravenous injection of radiopharmaceutical agent, a whole-body scintigraphy was acquired. The dosage of radiopharmaceutical agent ranged between 600 and 740 MBq depending on the body type of the patient, whereas the common intravenous injection is 670 MBq.

After bone scintigraphy, 586 planar bone scan images from patients with known P-Ca were reviewed, retrospectively. Anterior as well as posterior views of $1024 \times 256$-pixel resolution were digitally recorded using the whole-body field. The number of detected gamma decays in each spatial unit is represented by images of a 16-bit grayscale depth.

## 2.2. Main Aspects of Convolutional Neural Networks

One of the most powerful deep-learning techniques is the use of CNNs and they are common in analyzing and classifying images. Their architecture consists of the perceptron model, which means that it has fully connected layers, with each neuron connecting to every other neuron to the next layer. It is interesting to examine all the different types of layers that it contains.

First, there is the convolutional layer, whose name is similar to the name of the neural network. This specific type of layer is fundamental for CNNs, because it creates activation maps. That means that it learns the patterns of the image, helping the algorithm to classify any new image based on these patterns [78,79].

Second comes the pooling layer, which is very important for down-sampling the image and removing any noise that may fuzzy the algorithm. It works on a specific threshold, where it keeps any pixel value higher than the threshold and discards any value lower than the threshold.

The last layer of a CNN is the fully connected layer, which "flattens" the output of previous node, which in definition, means that it converts it into a vector, so it will be ready for an output. Following this, our algorithm gives a label to each image. There are subcategories to the fully connected layer, as the first one does the vectorization and the last one gives the probabilities for each of the given class. [80–82].

*2.3. Methodology*

In medical analysis, the use of CNNs is common, so we can identify with an automated algorithm whether a patient has bone cancer metastasis, or not. The CNN method for bone metastasis classification includes five processing stages (see Figure 2): obtain data from whole-body scans, data pre-processing to normalize the collected image data, CNN training, CNN validation and finally, CNN testing followed by the evaluation of the classification results, as presented in Section 4.3. The five stages of the proposed methodology are thoroughly presented in the following.

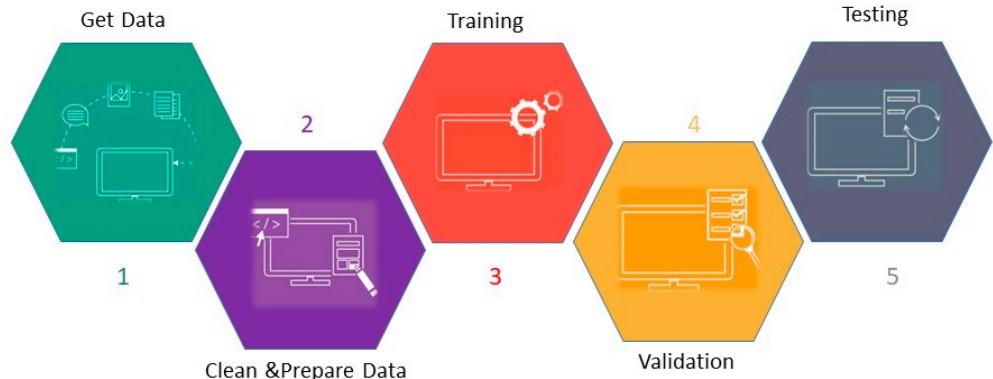

**Figure 2.** Five stages of the proposed methodology.

**Stage 1: Obtain Data**

The whole-body scans given by the nuclear medicine physician were in RGB (red, green, blue) mode and were stored in the PC memory. Depending on their class, whether they were normal or malignant, we assigned to them the equivalent prefix. This technique helped us to identify them inside the deep-learning algorithm and to pass them the correct output. Specifically we gave 0 to malignant images as output and 1 to benign ones.

**Stage 2: Clean and Prepare Data**

Step 2.1: Data normalization

Data Normalization is used in every machine learning process. This procedure rescales the values of the data within the range of 0 and 1, while it is important to know or estimate accurately, the minimum and maximum observable values. The Min-Max normalization method has been selected as the most popular method, as regards the performance of the systems being examined. Moreover, reviewing the relevant literature, studies showed that the results' accuracy from this method is higher than of those being produced by other normalization methods [83]. The scale that all images are within zero and one also prevents having outliers which will confuse the algorithm.

Step 2.2: Data shuffle

From splitting the dataset into training and testing emerges the danger of choosing wrong samples for our algorithm, and so the solution to this problem is provided by the application of shuffling method. That gives a random order to the data.

Step 2.3: Data split

We split the available image data into three parts, which are training, validation and testing. Each partition has specific percentage, which had been taken from the whole dataset, meaning that 15% of the dataset was given to testing and 85% is for the other two parts, namely we split it 80% for training and 20% for validation. Therefore, three different datasets are created: Training, Validation and Testing datasets, respectively.

**Stage 3: Training**

Step 3.1: Data augmentation

The method of data augmentation is usually used when we have a small number of data and we do not want to create a model that does not have generalization capabilities. For instance, with

the techniques of rescale, rotation range, zoom range and flip, we create variant images and we avoid overfitting.

Step 3.2: Define CNN architecture

To define a proper architecture of a CNN for image classification, an exploration process is followed. In the experimentation and trial phase, the most important parameters that can lead to effective network architecture were explored; the number and the type of convolutional layers, the number of nodes, the number of pooling layers, the filters, the drop rate and the batch size, the number of dense nodes.

Step 3.3: Define the functions of the CNN architecture

After a search in the relevant literature, the main functions of the CNN architecture are the activation function which is the function that defines the output of the layer, and the loss function which is the function that is used for the optimization of the network's weights. After thorough experiments illustrated in previous works, ReLU (rectified linear unit) is the most used activation function for convolutional layers and dense layers. As regards the output layer, Sigmoid is one of the most commonly used functions [78].

Step 3.4: Train CNN

As the name indicates, training a CNN means training the model with images offering the opportunity to learn patterns, as the more differentiated the training is, the better the model becomes. The training process begins when the algorithm tries to create a function that describes the desired relation, based on the training data. Afterwards, it makes predictions based on this function (it is an error function) and moves to the validation step.

**Stage 4: Validation**

The validation dataset is used exclusively for the validation process. During validation, the weights are normalized, and the algorithm makes predictions on already known data. During the validation process, the main aim is to minimize the error which reveals the prediction efficiency of the proposed method.

**Stage 5: Testing/Evaluation**

After the completion of the training and validation phases, the best learned model is deployed for the testing/evaluation process [84,85]. The evaluation process for the CNN model is accomplished using the testing data, as initially split, which is completely unknown to the model. During the testing phase, we check whether our classifier can make correct labeling to bot familiar images thus assessing the predictive ability of the current model. The classifier makes predictions on each image's class and finally compares the calculated predicted class to the true class. Next, for classifier evaluation, some well-known and popular performance metrics, such as the testing accuracy, precision, recall, sensitivity, specificity and f1-score of the model are computed, and accompanied by employing an error/confusion matrix for further model evaluation [84,86,87].

## 3. The Proposed CNN Architecture

The main goal of this research study is to create a deep neural network, and particularly, a CNN to precisely identify bone metastasis from whole-body scans, in women suffering from breast cancer. The developed CNN, which was applied in bone scintigraphy to automatically identify bone metastasis, will prove its ability to provide high accuracy with low computation time for whole-body image prediction. In the CNN exploration process, we had to experiment with different values for our parameters like pixels, epochs, drop rate, batch size, number of nodes and layers [79,80,88,89].

After a large number of experiments conducted with the aforementioned values, the best CNN architecture was derived. Actually, the best architecture consists of three convolutional layers followed by max pooling and dropout layer every time. Following the above layers, flatten comes next, along with the dropout layer. Then one dense layer is shown, and, in the end, the output layer takes over (see Figure 3). Table 1 summarizes all the information about the relevant architecture.

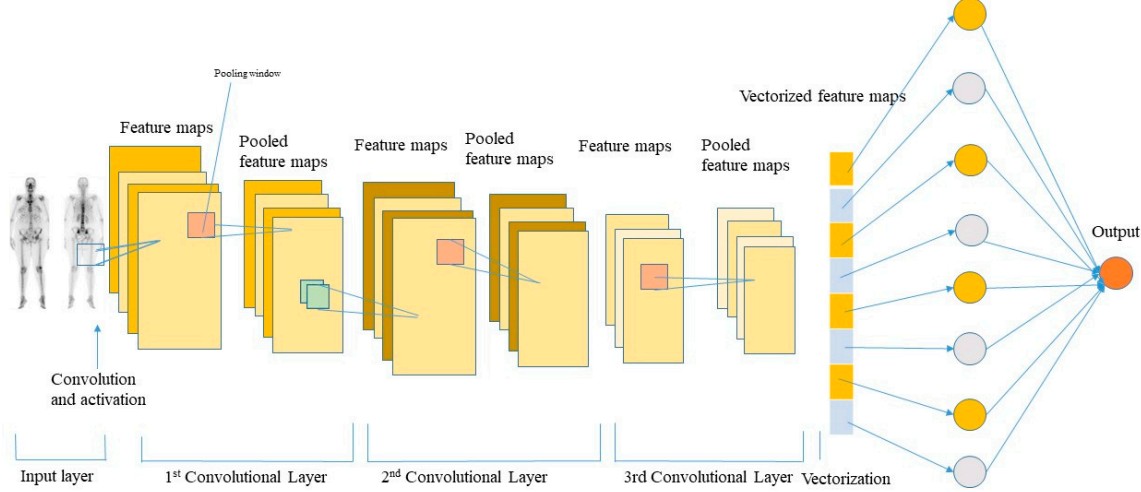

**Figure 3.** The CNN framework for bone scintigraphy classification.

The proposed CNN architecture has an image input size of 256 × 256 pixels, and we pass the images in RGB mode. Three convolutional layers are present, where the max pooling and the dropout layers follow each one of them. It needs to be highlighted that all convolutional layers use ReLU as activation function and the values of filters are doubled for every convolutional layer. This means that the first convolutional layer has 8 filters, the second layer has 16 layers and so on, while all of them have a 3 × 3 pixels kernel size. Furthermore, all pooling layers retain their max values and have 2 × 2 pixels size. In addition, dropout layers have the value 0.7 as their drop rate parameter. Next, the flatten layer follows to convert the data into one dimension; hence, the data are ready for the fully connected layer [90].

In the next phase, the dropout layer takes over, having the same value for drop rate, and helps the model to avoid overfitting. Next, the fully connected layer takes control of the data, comprising 64 nodes and ReLU as activation function. Finally, yet essentially, there is the output layer which has one (1) node as it deals with binary classification and contains the Sigmoid as activation function.

In the following figure (Figure 4), the proposed CNN architecture is illustrated, which produces the best results distinguishing whether a person has bone metastasis or not. We concluded to this architecture after experimenting with different number of epochs, pixels, drop rates, number of nodes, number of layers, and so we infer that it performs in a perfect way, having reached a very high accuracy with low value of loss.

| Layers | Output | Description |
|---|---|---|
| conv2d_1 | 254 x 254 | Kernel: 3 x 3, Filters : 8, ReLU |
| max_pooling2d_1 | 127 x 127 | 2 x 2 max pooling |
| dropout_1 | 127 x 127 | Drop rate = 0.7 |
| conv2d_2 | 125 x 125 | Kernel: 3 x 3, Filters : 16, ReLU |
| max_pooling2d_2 | 62 x 62 | 2 x 2 max pooling |
| dropout_2 | 62 x 62 | Drop rate = 0.7 |
| conv2d_3 | 60 x 60 | Kernel: 3 x 3, Filters : 32, ReLU |
| max_pooling2d_3 | 30 x 30 | 2 x 2 max pooling |
| dropout_3 | 30 x 30 | Drop rate = 0.7 |
| Flatten_1 | 28800 | |
| dropout_4 | 28800 | Drop rate = 0.7 |
| dense_1 | 64 | Nodes: 64, ReLU |
| dense_2 | 1 | Nodes: 1, Sigmoid |

**Figure 4.** The selected CNN architecture for diagnosis of metastatic breast cancer in bones.

## 4. Results

In this section, we have gathered the most representative and useful results of the conducted experiments and CNN exploration, after 10 executions. The final results are the averaged values of those produced from the experiments after each experiment was performed for 10 times.

This experiment was performed in Colaboratory, called Google Colab [91], which is a free Jupyter notebook environment in the cloud. The main reason for selecting this cloud environment of Google Colab is that supports free GPU. The frameworks Keras 2.0.2 and TensorFlow 2.0.0. were used, python language 3.7, CNN with structure (Convolution layer, 3; Maxpooling layer, 3) and Adam Optimizer.

Other CNN methods, including VGG16 [33], ResNet50 [34], DenseNet121 [35], and MobileNet [92], were also implemented so as a comparative analysis could be performed between them and the method proposed in this paper.

### 4.1. Classification Performance Evaluation

In the evaluation phase, we use specific metrics to see how our algorithm worked with the given images. The validation metrics that were used are accuracy, precision, recall, F1 score, sensitivity and specificity.

A predicted sample can be classified in one of the four states in binary classification problems: true positive (TP), true negative (TN), false positive (FP), and false negative (FN). True positive means that the label of the image is benign, and it is classified correctly. False positive means that the label of the image is not benign, and it is classified as benign. True negative means that the label of the image is not benign, and it is classified as not benign. Similarly, false negative means that the label of the image is benign, and it is classified as malignant.

Mathematical formulations of the six performance metrics are defined as follows:

$$Accuracy = \frac{TP + TN}{TP + FP + TN + FN} \tag{1}$$

$$Precision = \frac{TP}{TP + FP} \tag{2}$$

$$Recall = \frac{TP}{TP + TN} \tag{3}$$

$$F1 - score = \frac{2 * (Recall \times Precision)}{Recall + Precision} \tag{4}$$

$$Sensitivity = \frac{TP}{TP + FN} \tag{5}$$

$$Specificity = \frac{TN}{TN + FP} \tag{6}$$

Accuracy is the ratio between the correct labeled samples and all samples. Precision points out the ratio of corrected images of positive class to all images that were predicted for the same category. Recall deals with the division of corrected positive images to all corrected images of both categories.

F1-score is a harmonic mean between recall and precision. Sensitivity is the proportion of images which test positive for the malignancy (bone metastasis) among those which have the metastasis/malignancy. Finally, we have specificity, which is the balance between the corrected negative images and the sum of numerator and wrong labeled positive images.

### 4.2. Confusion Matrix

It is common to use a confusion matrix in machine-learning problems because it is an efficient way to observe the performance of an algorithm, which in our case is image classification. It is a useful evaluation metric for calculating the errors made by the classifier and gives precisely the number of images that were classified at the wrong label. In this study the best confusion matrix of different CNN architectures is produced with the least classification errors.

### 4.3. Presentation of Results

OpenCV was used for loading and manipulating images, Glob for reading filenames from a folder, Matplotlib for plot visualizations and finally Numpy for all mathematical and array operations. Python was used for coding, with the CNN being programmed with Keras (with Tensorflow [92]), the data normalization, data splitting, confusion matrices and classification reports with Sci-Kit Learn. The computations ranged between 2′ to 4′ per training for grayscale images and 2′ to 5′ per training (epoch) for RGB images (256 × 256 pixels), depending on the different input, with the grayscale images being the fastest and RGB images being the slowest to train.

To accomplish the aim of this study, we initially employed different CNN-based architectures and hyperparameter selection as defined in Section 3. After a thorough CNN exploration analysis, we concluded that the best CNN configuration was the following: a CNN with 3 convolutional layers, starting from 8 filters for the first layer, and for each convolutional layer that comes next, the number of filters is doubled (8->16->32), just like the following max-pooling layers. All of those filters have dimensions of 3 × 3. This set-up is shown in Table 1.

Following the steps described in Section 2.2, first we load the images and select the image type, RGB or grayscale. The images are in RGB mode by default and only after requesting are they transformed to grayscale mode. The second and important step is to normalize them, because we want to exclude any outliers and fuzzy data from our dataset. Furthermore, we should shuffle to randomize the order and then perform data augmentation to create variance to our data. Following that, we split our dataset into training, validation and testing and then we pass our images through the CNN. The proposed CNN network has been pre-trained on the ImageNet data set [93]. This data set has proven to be of high utility as it can provide another, more efficient method of weights initialization, in any image-related task.

We followed a two-class classification problem, for bone metastasis presence and absence in patients suffering from breast cancer, considering in total 408 samples of women patients.

Through our experimental analysis, working initially in RGB mode, we conducted experiments with various drop rates, epochs, number of dense nodes, pixel sizes and batch sizes. Different values for pixels were examined, such as 100 × 100, 200 × 200, 256 × 256, 300 × 300 and various values for batch sizes such as 8, 16 and 32 were investigated. In addition, variant drop rate values were studied, for example 0.2, 0.4, 0.7 and 0.9 and a divergent number of dense nodes like 16, 32 and 64 were explored. Epochs were explored from 100, 200, 300 and 500.

It is important to mention that we accomplished more runs with different epochs and pixels and we observed that for epochs = 200, for pixels (256,256), batch size 8, drop rate 0.7 and 64 nodes in dense layer, the CNN models worked better concerning the classification accuracy. In Appendix A, we provide some example simulations for different pixel sizes by also modifying the drop rate and batch size. Also, in Appendix A, some indicative results with different pixel sizes (100, 100), (200, 200) and (300, 300) are illustrated in Tables A1–A3 respectively.

After careful selection of number of epochs and pixels size (epochs = 200 and pixel = 256 × 256), we conducted further experiments with varying drop rates and batch sizes, to find out the best performance parameters.

The following Tables gather the performance analysis and results for the various CNN models with epochs = 200 and pixel = 256 × 256 (in Google Colab [91]).

After a thorough CNN exploration, the best CNN model has the following characteristics: batch-size = 8, dropout = 0.7, nodes (3 conv layers) = 8, 16, 32, Dense Nodes: 64, Epochs: 200 and Pixel = (256,256, 3). Figure 5 represents the accuracy and loss precision curves for the best CNN model.

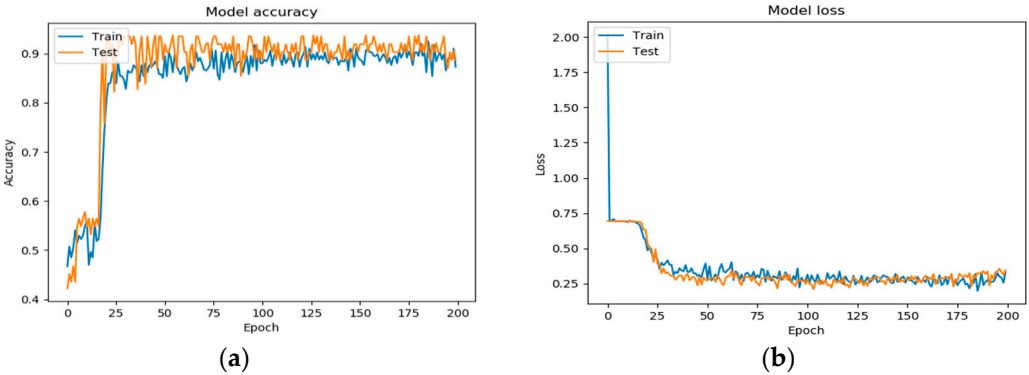

**Figure 5.** Precision curves for best CNN in RGB mode: (**a**) Accuracy and (**b**) Loss.

From the following Tables (see Tables 3–6), it is concluded that the best CNN model is the model with dropout = 0.7 and batch size = 8. The average time for 200 epochs is 788 s.

**Table 3.** CNN model 1 (epochs = 200, dropout = 0.2, pixel = 256 × 256) runs for different batch sizes.

|  | **Batch Size = 8** | | | | **Batch Size = 16** | | | | **Batch Size = 32** | | | |
|---|---|---|---|---|---|---|---|---|---|---|---|---|
|  | Acc. Val | Loss Val | Acc Test | Loss Test | Acc. Val | Loss Val | Acc Test | Loss Test | Acc. Val | Loss Val | Acc Test | Loss Test |
| Run 1 | 85.94 | 0.22 | 91.1 | 0.27 | 84.37 | 0.47 | 89.58 | 0.35 | 98.06 | 0.23 | 87.5 | 0.17 |
| Run 2 | 79.69 | 0.65 | 75 | 1.1 | 84.37 | 0.45 | 79.17 | 0.54 | 92.18 | 0.18 | 84.37 | 0.28 |
| Run 3 | 87.5 | 0.3 | 98.21 | 0.1 | 87.5 | 0.58 | 85.42 | 0.41 | 82.81 | 0.28 | 81.25 | 0.39 |
| Run 4 | 89.1 | 0.49 | 92.86 | 0.38 | 92.18 | 0.21 | 85.42 | 0.33 | 85.93 | 0.34 | 90.62 | 0.49 |
| Run 5 | 92.19 | 0.34 | 82.15 | 0.6 | 93.75 | 0.21 | 91.7 | 0.27 | 98.43 | 0.008 | 90.62 | 0.18 |
| Average | **86.89** | **0.4** | **87.86** | **0.49** | **88.43** | **0.38** | **86.26** | **0.38** | **91.48** | **0.207** | **86.87** | **0.305** |

**Table 4.** CNN model 2 (epochs = 200, dropout = 0.7, pixel = 256 × 256) runs for different batch sizes.

|  | **Batch Size = 8** | | | | **Batch Size = 16** | | | | **Batch Size = 32** | | | |
|---|---|---|---|---|---|---|---|---|---|---|---|---|
|  | Acc. Val | Loss Val | Acc Test | Loss Test | Acc. Val | Loss Val | Acc Test | Loss Test | Acc. Val | Loss Val | Acc Test | Loss Test |
| Run 1 | 93.75 | 0.21 | 89.28 | 0.37 | 93.75 | 0.15 | 83.33 | 0.37 | 89.06 | 0.42 | 93.75 | 0.207 |
| Run 2 | 95.31 | 0.35 | 94.64 | 0.26 | 90.62 | 0.29 | 89.58 | 0.23 | 95.31 | 0.16 | 87.5 | 0.33 |
| Run 3 | 92.18 | 0.23 | 92.86 | 0.18 | 95.31 | 0.16 | 93.75 | 0.22 | 90.62 | 0.22 | 93.75 | 0.18 |
| Run 4 | 93.75 | 0.24 | 92.85 | 0.19 | 89.1 | 0.29 | 89.58 | 0.22 | 95.31 | 0.15 | 87.5 | 0.314 |
| Run 5 | 87.5 | 0.29 | 92.85 | 0.24 | 90.62 | 0.23 | 87.51 | 0.26 | 92.18 | 0.24 | 93.75 | 0.239 |
| Run 6 | 89.06 | 0.3 | 89.28 | 0.22 | 89.1 | 0.29 | 97.92 | 0.14 | 82.81 | 0.35 | 90.62 | 0.227 |
| Run 7 | 90.62 | 0.287 | 94.64 | 0.21 | 84.38 | 0.32 | 95.83 | 0.2 | 98.43 | 0.16 | 93.75 | 0.21 |
| Run 8 | 92.18 | 0.22 | 87.5 | 0.25 | 92.18 | 0.23 | 93.75 | 0.19 | 85.93 | 0.28 | 87.5 | 0.232 |
| Run 9 | 84.37 | 0.33 | 96.42 | 0.16 | 85.7 | 0.32 | 95.83 | 0.24 | 90.62 | 0.21 | 93.75 | 0.19 |
| Run 10 | 93.75 | 0.176 | 94.64 | 0.154 | 89.06 | 0.37 | 87.5 | 0.26 | 85.93 | 0.39 | 87.5 | 0.32 |
| Average | **91.25** | **0.263** | **92.49** | **0.224** | 89.98 | 0.27 | 91.46 | 0.23 | 90.62 | 0.26 | 90.94 | 0.25 |

**Table 5.** CNN model (epochs = 200, pixel = 256 × 256, conv (8->16->32), dense = 64) runs for different dropouts.

| **Batch Size = 8** | **Dropout = 0.2** | | | | **Dropout = 0.4** | | | | **Dropout = 0.9** | | | |
|---|---|---|---|---|---|---|---|---|---|---|---|---|
|  | Acc. Val | Loss Val | Acc Test | Loss Test | Acc. Val | Loss Val | Acc Test | Loss Test | Acc. Val | Loss Val | Acc Test | Loss Test |
| Run 1 | 85.94 | 0.22 | 91.1 | 0.27 | 93.75 | 0.22 | 93.75 | 0.23 | 95.31 | 0.31 | 85.71 | 0.43 |
| Run 2 | 79.69 | 0.65 | 75 | 1.1 | 87.5 | 0.42 | 85.71 | 0.32 | 82.81 | 0.37 | 82.14 | 0.38 |
| Run 3 | 87.5 | 0.3 | 98.21 | 0.1 | 84.37 | 0.34 | 89.28 | 0.23 | 79.69 | 0.47 | 73.21 | 0.48 |
| Run 4 | 89.1 | 0.49 | 92.86 | 0.38 | 85.94 | 0.29 | 89.28 | 0.23 | 87.5 | 0.36 | 80.36 | 0.43 |
| Run 5 | 92.19 | 0.34 | 82.15 | 0.6 | 85.94 | 0.23 | 83.93 | 0.39 | 84.37 | 0.36 | 89.28 | 0.27 |
| Average | **86.89** | **0.4** | **87.86** | **0.49** | **87.52** | **0.32** | **87.68** | **0.3** | **85.94** | **0.37** | **82.14** | **0.4** |

**Table 6.** CNN model (epochs = 200, pixel = 256 × 256, conv (8->16->32), dropout = 0.7) runs for different dense nodes.

| **Batch size = 8** | **Dense Nodes = 16** | | | | **Dense Nodes = 32** | | | | **Dense Nodes = 64** | | | |
|---|---|---|---|---|---|---|---|---|---|---|---|---|
|  | Acc. Val | Loss Val | Acc Test | Loss Test | Acc. Val | Loss Val | Acc Test | Loss Test | Acc. Val | Loss Val | Acc Test | Loss Test |
| Run 1 | 95.31 | 0.19 | 91.1 | 0.18 | 85.94 | 0.39 | 91.1 | 0.23 | 93.75 | 0.21 | 89.28 | 0.37 |
| Run 2 | 95.31 | 0.23 | 91.1 | 0.22 | 92.19 | 0.18 | 85.71 | 0.25 | 95.31 | 0.35 | 94.64 | 0.26 |
| Run 3 | 85.94 | 0.29 | 87.5 | 0.32 | 92.19 | 0.25 | 87.50 | 0.27 | 92.18 | 0.237 | 92.86 | 0.18 |
| Run 4 | 95.31 | 0.17 | 91.1 | 0.23 | 90.62 | 0.23 | 91.10 | 0.27 | 93.75 | 0.238 | 92.86 | 0.19 |
| Run 5 | 87.5 | 0.27 | 92.86 | 0.16 | 90.62 | 0.22 | 94.64 | 0.21 | 87.5 | 0.29 | 92.85 | 0.24 |
| Average | **91.87** | **0.23** | **90.73** | **0.22** | **90.31** | **0.25** | **90.01** | **0.25** | **92.50** | **0.27** | **92.50** | **0.25** |

Table 7 depicts the performance metrics for both malignant (metastasis) and benign (no metastasis) patients for this network configuration. The best confusion matrix is illustrated in Table 8.

To further investigate the performance of the proposed CNN architecture, the grayscale mode was applied, for various CNN parameters with respect to drop-rate, batch size, dense nodes, pixel sizes and number of epochs. The same CNN exploration as in RGB mode was followed. After a thorough CNN exploration for grayscale mode, the best network parameters were selected: 3 convolutional layers (2, 4, 8), dense = 16, pixel size 150 × 150 and drop-rate = 0.2, epochs = 200, batch size = 8. The results for the best network performance for the bone metastasis classification problem, concerning the grayscale mode images, are gathered in Table 9. Table 10 depicts the average values of the five performance metrics and the best produced confusion matrix for grayscale mode. Figure 6 illustrates the plots of precision curves concerning accuracy and loss for the best CNN in grayscale mode. The best time was Best: 196 s, Average: 205 s, Steps: 28–30 step. Also, Figure 7 depicts the average prediction errors for the examined CNN architectures in RGB and Grayscale mode, showing the outperformance of the best CNN architecture with dropout=0.7.

**Table 7.** Performance metrics for the best CNN for 10 runs (batch size = 8, dropout = 0.7).

| Runs | | Precision | Recall | F1-Score | Sensitivity | Specificity |
|---|---|---|---|---|---|---|
| Run 1 | Malignant | 0.94 | 0.89 | 0.92 | 0.89 | 0.92 |
| | Benign | 0.85 | 0.92 | 0.88 | 0.85 | 0.94 |
| Run 2 | Malignant | 0.93 | 1.00 | 0.96 | 1.00 | 0.88 |
| | Benign | 1.00 | 0.88 | 0.93 | 0.88 | 1.00 |
| Run 3 | Malignant | 0.9 | 1.00 | 0.95 | 1.00 | 0.85 |
| | Benign | 1.00 | 0.85 | 0.92 | 0.85 | 1.00 |
| Run 4 | Malignant | 1.00 | 0.88 | 0.93 | 0.88 | 1.00 |
| | Benign | 0.88 | 1.00 | 0.94 | 1.00 | 0.88 |
| Run 5 | Malignant | 1.00 | 0.88 | 0.94 | 0.88 | 1.00 |
| | Benign | 0.88 | 1.00 | 0.94 | 1.00 | 0.88 |
| Run 6 | Malignant | 0.87 | 0.93 | 0.9 | 0.93 | 0.88 |
| | Benign | 0.94 | 0.88 | 0.91 | 0.88 | 0.93 |
| Run 7 | Malignant | 0.97 | 0.93 | 0.95 | 0.93 | 0.97 |
| | Benign | 0.94 | 0.97 | 0.95 | 0.97 | 0.93 |
| Run 8 | Malignant | 0.82 | 0.96 | 0.89 | 0.96 | 0.82 |
| | Benign | 0.97 | 0.82 | 0.89 | 0.82 | 0.96 |
| Run 9 | Malignant | 0.97 | 0.94 | 0.96 | 0.94 | 0.96 |
| | Benign | 0.93 | 0.96 | 0.95 | 0.96 | 0.94 |
| Run 10 | Malignant | 0.94 | 0.97 | 0.96 | 0.97 | 0.93 |
| | Benign | 0.96 | 0.93 | 0.94 | 0.93 | 0.97 |
| **Average** | Malignant | 0.934 | 0.938 | 0.936 | 0.94 | 0.92 |
| | Benign | 0.935 | 0.921 | 0.925 | 0.91 | 0.94 |

**Table 8.** Best confusion matrix for the proposed CNN.

| Title 1 | Malignant | Benign |
|---|---|---|
| Malignant | 31 | 1 |
| Benign | 2 | 25 |

**Table 9.** CNN performance for grayscale images.

| Grayscale | CNN Architecture (2,4,8), Epochs = 200, Batch Size = 8, Dense = 16, Drop-Rate = 0.2, Pixel Size (150 × 150) | | | |
|---|---|---|---|---|
| Runs | Accuracy (Validation) | Loss (Validation) | Accuracy (Testing) | Loss (Testing) |
| Run 1 | 85.94 | 0.31 | 85.71 | 0.42 |
| Run 2 | 92.19 | 0.28 | 80.36 | 0.45 |
| Run 3 | 78.12 | 0.52 | 75 | 0.51 |
| Run 4 | 87.5 | 0.26 | 78.57 | 0.61 |
| Run 5 | 84.37 | 0.33 | 89.28 | 0.3 |
| Average | 85.62 | 0.34 | 81.78 | 0.46 |



**Table 10.** Performance metrics and confusion matrix for grayscale images.

| Grayscale | Precision | Recall | F1-score | Sensitivity | Specificity |
|---|---|---|---|---|---|
| Malignant | 0.82 | 0.97 | 0.89 | 0.82 | 0.96 |
| Benign | 0.96 | 0.76 | 0.85 | 0.96 | 0.82 |
| **Confusion matrix** | **Malignant** | **Benign** | | | |
| Malignant | 32 | 1 | | | |
| Benign | 7 | 22 | | | |

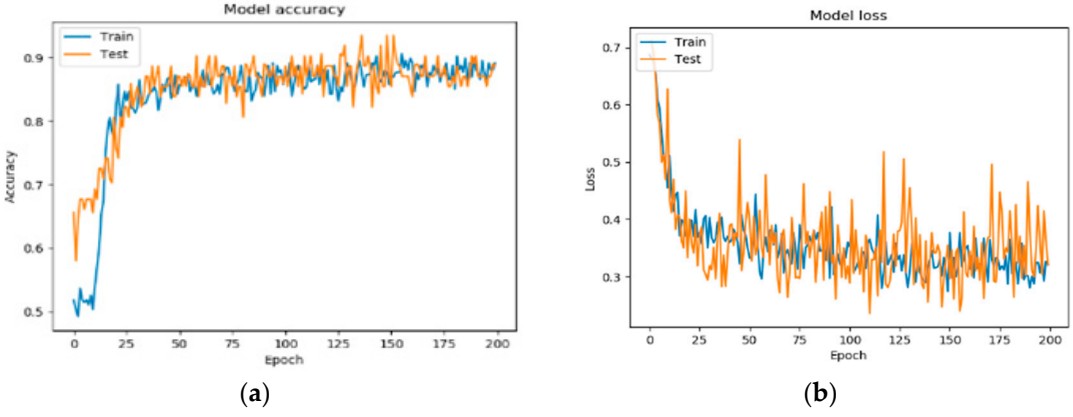

(**a**) (**b**)

**Figure 6.** Precision curves for best CNN in grayscale mode: (**a**) accuracy and (**b**) loss.

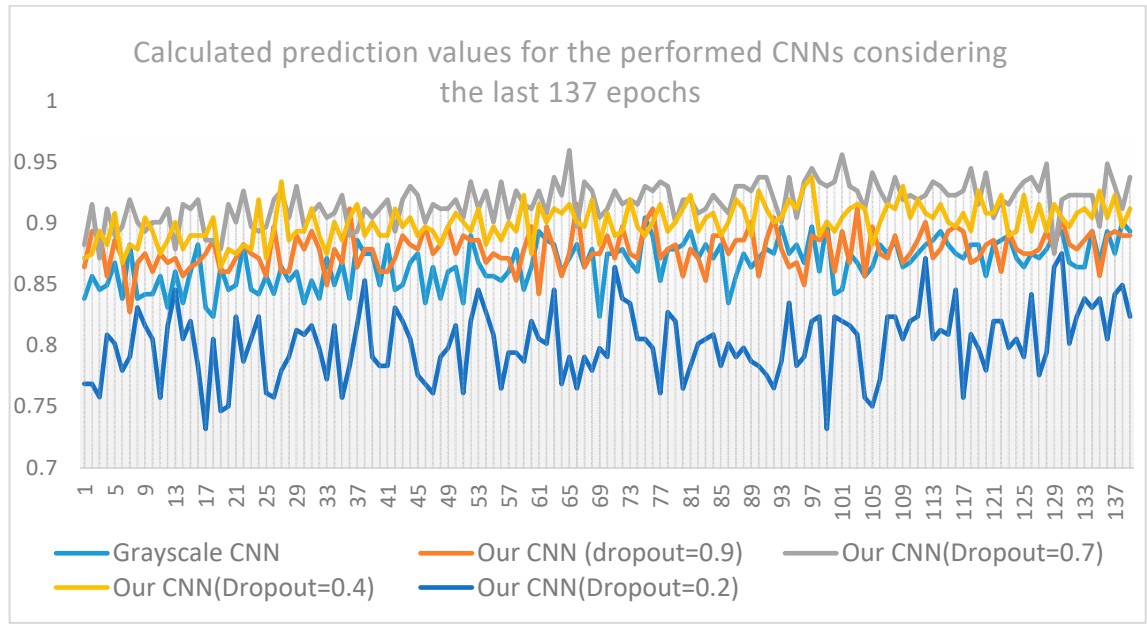

**Figure 7.** Average prediction errors for the best CNN architecture.

### 4.4. Comparison with Benchmark CNNs

In what follows, an extensive comparative analysis between the state-of-the-art CNNs such as VGG16 [33], ResNet50 [34], Mobile Net [92] and Densenet [35] and our best model was performed. The following well-known CNN architectures were used: (i) ResNet50, is a 50-weight layer deep version of ResNet (Residual neural Network), with 152 layers, based on "network-in-network" micro-architectures [94]. ResNet has less parameters than the VGG network, demonstrating that extremely deep networks can be trained using standard SGD (Stochastic gradient descent) and a reasonable initialization function through the use of residual modules. (ii) VGG16 [33], is an extended

version of VGG as it contains 16 weight layers within the architecture. VGGs are usually constructed by using $3 \times 3$ convolutional layers which are stacked on top of each other.

DenseNET is an extension of ResNet [95]. Based on recent research, it has been proven that using Neural Networks which have smaller connections between layers, mainly in the case where the dataset is relatively small, give better classification performance. Based on this result, DenseNet was created. It follows the feed-forward technique and each layer has as the input all previous output feature maps [35].

MobileNet convolutes each channel separately instead of combining and flattening them all, with the use of depthwise separable convolutions [92]. Its architecture combines convolutional layers, depthwise and pointwise layers to a total number of 30.

These popular CNNs were used for transfer learning and were used with weights that were trained on the ImageNet [93] dataset. Tables 11–13 gather the results of these benchmark CNN models compared with our best CNN configuration.

**Table 11.** Accuracy and loss of the proposed CNN model and benchmark CNNs.

| Average | Best CNN | ResNet50 | VGG16 | MobileNet | DenseNet |
|---|---|---|---|---|---|
| Acc. Validation | 92.50 | 91.87 | 82.49 | 88.13 | 93.75 |
| Loss Validation | 0.27 | 0.24 | 0.37 | 0.25 | 0.15 |
| Acc. Testing | 92.50 | 91.87 | 83.75 | 85.36 | 95 |
| Loss Testing | 0.25 | 0.19 | 0.34 | 0.35 | 0.176 |

**Table 12.** Malignant disease class performance comparison of different methods.

| | Best CNN | ResNet50 | VGG16 | MobileNet | DenseNet |
|---|---|---|---|---|---|
| Accuracy | 92.50 | 91.87 | 83.75 | 85.36 | 95 |
| Precision | 0.93 | 0.84 | 0.85 | 0.78 | 0.92 |
| Recall | 0.94 | 1 | 0.9 | 0.99 | 0.99 |
| F1-score | 0.94 | 0.91 | 0.86 | 0.87 | 0.95 |
| Sensitivity | 0.94 | 1 | 0.89 | 0.99 | 0.99 |
| Specificity | 0.92 | 0.78 | 0.75 | 0.66 | 0.9 |

**Table 13.** Benign class performance comparison of different methods.

| | Best CNN | ResNet50 | VGG16 | MobileNet | DenseNet |
|---|---|---|---|---|---|
| Accuracy | 92.50 | 91.87 | 83.75 | 85.36 | 95 |
| Precision | 0.94 | 1 | 0.87 | 0.99 | 0.98 |
| Recall | 0.92 | 0.78 | 0.75 | 0.66 | 0.9 |
| F1-score | 0.93 | 0.87 | 0.77 | 0.79 | 0.94 |
| Sensitivity | 0.91 | 0.78 | 0.75 | 0.66 | 0.91 |
| Specificity | 0.94 | 1 | 0.89 | 0.99 | 0.99 |

In this research work, after an extensive exploration with the provided architectures of popular CNNs, the following parameters for the well-known CNNs were defined.

- Best CNN: batch-size = 8, dropout = 0.7, nodes (3 conv layers) = 8,16,32, dense nodes: 64, epochs:200, pixel = (256,256,3)
- VGG16: pixel size ($200 \times 200$), batch size = 32, dropout = 0.2, dense nodes $2 \times 512$, epochs = 200
- ResNet50: pixel size ($224 \times 224$), batch size = 32, dropout = 0.5, dense nodes $32 \times 32$, epochs = 200
- MobileNet: pixel size ($250 \times 250$), batch size = 8, dropout = 0.5, global average pooling, dense = $1500 \times 1500$, epochs = 200
- DenseNet: pixel size (200X200), batch size = 8, dropout = 0.9, global average pooling, dense nodes = ($128 \times 128$), epochs = 200.

In Appendix B, Figures A1–A4 illustrate the average prediction accuracies with their respective loss curves for the optimal architectures of the four benchmark CNNs. The average running time for each CNN architecture for all models was calculated and Figure 8 depicts the values of running time for 200 epochs. It is obvious that the proposed CNN model has similar average run time to the other popular CNN models.

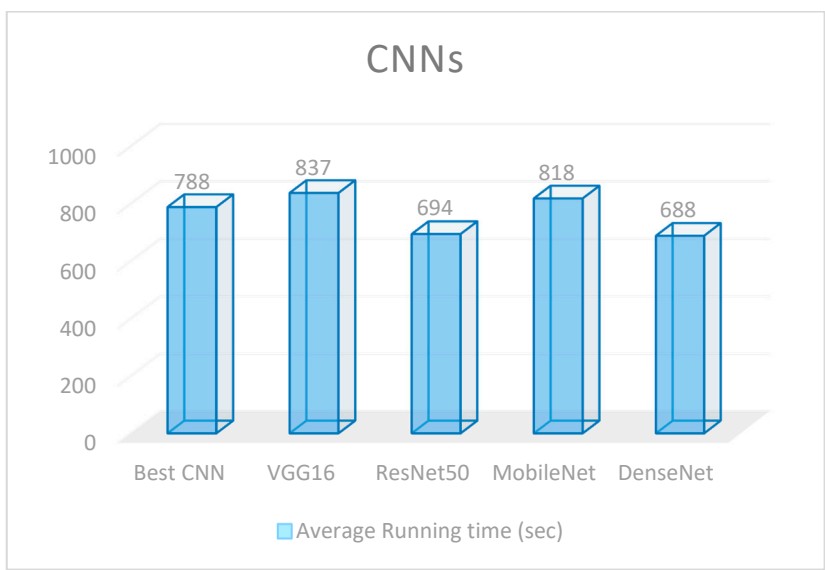

**Figure 8.** Average run time for the compared best CNN architectures.

## 5. Discussion

Through the trials and experiments performed, it emerges that the accuracy of the selected CNN architecture is higher in the case of RGB images. Also, the loss (validation and testing) is quite low, and the confusion matrix is better in terms of accuracy than the corresponding one in the case of grayscale images, showing the superiority of RGB images use for the classification task. The CNN in RGB process spent 4 min approximately for training each fold dataset and less than 30 s for prediction.

The best performing CNN algorithm was then compared with some state-of-the art CNNs that are commonly used for image classification problems. To directly compare the classification performance of our model with other popular CNN architectures we used, besides the accuracy, the evaluation metrics of precision, recall, F1 score, sensitivity and specificity indicators, which are listed in the tables above. We highlight that our proposed algorithm performed better when compared to popular CNN algorithms for the specific problem. This shows that the improvements made to network architecture and the configuration of hyperparameter were rather effective. This study validates the premise that CNNs are algorithms that can offer high accuracy in medical image classification-based problems. This can have a direct application on medical imaging where the automatic identification of diseases is crucial for the patients.

The main outcomes of this study can be summarized as follows:

■ The proposed CNN method exhibits outstanding performance when RGB analysis is performed, for the examined images of this case study. This can emanate from the fact that the results produced from the application of the CNN architecture are based on distinct features that appear specifically on the bone metastasis presence scans, compared to the healthy scans (no malignant spots).

■ The proposed CNN architecture is superior compared to three out of four benchmarks and well-known CNN architectures (ResNet50, VGG16, MobileNet, and DenseNet) which have been previously and efficiently used in medical image-processing problems. As observed from the aforementioned tables and figures, the results deriving from the application of the CNN method,

are better in terms of classification accuracy, prediction, sensitivity, specificity and f1 score, than those coming from popular and well-known CNN approaches found in the relevant literature. Even if CNN has an inferior performance compared to DenseNet, the accuracy and loss values of the proposed CNN show a quite low variability whereas the corresponding DenseNet values present substantial variations among predicted outcomes (see Figure A3).

- ■ The proposed bone scan deep learning performs efficiently despite the fact that it was trained on a small number of images.
- ■ Overall, the deployed process seemingly improves the diagnostic effect of the deep-learning method, making it more efficient compared to other benchmark CNN architectures for image analysis. For the purpose of medical image analysis and classification, the CNN approach can be used effectively in the classification of whole-body images in nuclear medicine, outweighing the popular CNN architectures for medical image analysis.

To sum up, this is the first research study concerning the application of CNNs efficiently in bone metastasis diagnosis from breast cancer patients in bone scintigraphy. Following the proposed methodology, a robust CNN algorithm is created for bone metastasis diagnosis using whole-body scans from breast cancer patients. This algorithm can be implemented in any system without specific requirements, providing the nuclear medicine physician with an efficient and robust decision support tool to automatically identify whether a patient is suffering from breast cancer or not, by looking at whole-body scintigraphy images. Due to its high classification accuracy and performance, this algorithm has a significant positive impact in clinical diagnosis and decision making in nuclear medicine by providing diagnosis to metastatic breast cancer patients.

The proposed classification algorithm can be applied in any type of whole-body scans in bone scintigraphy and can be integrated into a CAD system of whole-body diagnosis or a clinical decision support system in nuclear medical imaging. Hence, it can be used by physicians in medical practice.

## 6. Conclusions

In this research study, convolutional neural network algorithms have been used to deal with the problem of bone metastasis diagnosis in breast cancer patients due to their significant applicability in image analysis. A total of 408 images were acquired with similar numbers of both benign and malignant cases (metastasis present). Two different types of processing images, RGB-mode and grayscale-mode, denoting their colored or monochromatic tone, were tested in order to evaluate their contribution to the increase of performance. Several CNN architectures were tested, and only one ended up with the best performance under all hyperparameter selection cases, with the results ranging from 85.35% to 93.75%.

In conclusion, these preliminary image data suggest that bone scintigraphy combined with the CNN method can have a considerable effect in the detection of bone metastasis. In particular, this approach allows an easier and more precise interpretation of the images that can have a positive impact on diagnosis accuracy as well as on decision making regarding the treatment that will be further administered.

Due to an insufficient amount of data in our study, a more detailed classification and diagnosis of breast cancer diseases needs to be carried out, and that is going to be our next priority in our future research. Since deep neural networks are based on complex, inter-connected hierarchical representations of the training data, which are used to make predictions, the task of interpreting these predictions becomes quite demanding. Moreover, due to the resemblance between deep neural networks and "blackbox" models, interpretability needs to be enhanced, while the research community should further investigate how to measure sensitivity and visualize features in deep learning.

In future work, the authors intend to show how the proposed algorithm could be integrated into a CAD system of whole-body diagnosis in nuclear medicine and also provide evidence that the clinical decision support system could be generalized across continents.

**Author Contributions:** Conceptualization, N.P. and E.P.; methodology, N.P., E.P. and A.A.; software, A.F. and A.A.; validation, N.P., E.P. and A.A.; formal analysis, N.P.; investigation, N.P.; resources, N.P.; data curation, N.P.; writing—original draft preparation, N.P.; writing—review and editing, N.P., and E.P.; visualization, A.F. and A.A.; supervision, N.P. and E.P.; project administration, E.P. All authors have read and agreed to the published version of the manuscript.

**Funding:** This research received no external funding.

**Acknowledgments:** 

**Conflicts of Interest:** The authors declare no conflict of interest.

## Appendix A

**Table A1.** CNNs (epochs = 200, dropout = 0.7, conv (8->16->32), dense = 64) runs for different pixel sizes.

| Batch size = 8 | CNN (100 × 100 × 3) | | | | CNN (200 × 200 × 3) | | | | CNN (300 × 300 × 3) | | | |
|---|---|---|---|---|---|---|---|---|---|---|---|---|
| | Acc. Val | Loss Val | Acc Test | Loss Test | Acc. Val | Loss Val | Acc Test | Loss Test | Acc. Val | Loss Val | Acc Test | Loss Test |
| Run 1 | 82.81 | 0.76 | 73.21 | 1.04 | 85.94 | 0.17 | 92.86 | 0.17 | 87.5 | 0.296 | 96.428 | 0.157 |
| Run 2 | 82.81 | 0.76 | 87.5 | 0.3 | 92.19 | 0.2 | 91.1 | 0.29 | 89.06 | 0.247 | 94,64 | 0.167 |
| Run 3 | 81.25 | 0.39 | 80.36 | 0.64 | 81.25 | 0.28 | 87.5 | 0.24 | 87.5 | 0.27 | 89.28 | 0.206 |
| Run 4 | 84.37 | 0.37 | 87.5 | 0.45 | 84.37 | 0.25 | 83.93 | 0.27 | 92.187 | 0.21 | 89.28 | 0.255 |
| Run 5 | 81.25 | 0.39 | 76.78 | 0.54 | 92.19 | 0.2 | 92.86 | 0.15 | 87.5 | 0.276 | 89.28 | 0.235 |
| Average | **82.5** | **0.53** | **81.07** | **0.59** | **87.19** | **0.22** | **89.65** | **0.22** | **88.75** | **0.26** | **91.78** | **0.20** |

**Table A2.** CNN Model (epochs = 200, pixel = 200 × 200, conv (8->16->32), dense = 64) runs for different dropouts.

| Batch size = 8 | Dropout = 0.2 | | | | Dropout = 0.4 | | | | Dropout = 0.9 | | | |
|---|---|---|---|---|---|---|---|---|---|---|---|---|
| | Acc. Val | Loss Val | Acc Test | Loss Test | Acc. Val | Loss Val | Acc Test | Loss Test | Acc. Val | Loss Val | Acc Test | Loss Test |
| Run 1 | 93.75 | 0.22 | 91.1 | 0.21 | 87.5 | 0.44 | 92.86 | 0.18 | 71.87 | 0.55 | 71.43 | 0.57 |
| Run 2 | 84.37 | 0.45 | 83.93 | 0.29 | 89.1 | 0.47 | 85.71 | 0.34 | 81.25 | 0.41 | 87.5 | 0.37 |
| Run 3 | 83.37 | 0.38 | 85.67 | 0.28 | 87.5 | 0.21 | 89.28 | 0.17 | 92.19 | 0.36 | 85.71 | 0.41 |
| Run 4 | 95.31 | 0.7 | 89.29 | 0.17 | 87.5 | 0.25 | 85.71 | 0.4 | 79.69 | 0.43 | 87.5 | 0.46 |
| Run 5 | 90.62 | 0.21 | 92.86 | 0.23 | 82.1 | 0.36 | 87.25 | 0.43 | 84.37 | 0.4 | 78.57 | 0.43 |
| Average | 89.48 | 0.39 | 85.57 | 0.24 | 86.74 | 0.34 | 88.16 | 0.3 | 81.87 | 0.43 | 82.14 | 0.45 |

**Table A3.** VGG16 (epochs = 200, dropout = 0.2, pixel = 256 × 256) runs for different batch sizes.

| | VGG16 (Batch Size=32) | | | |
|---|---|---|---|---|
| | Acc. Val | Loss Val | Acc Test | Loss Test |
| Run 1 | 84.37 | 0.3 | 81.25 | 0.36 |
| Run 2 | 73.44 | 0.53 | 78.12 | 0.26 |
| Run 3 | 75 | 0.46 | 84.37 | 0.38 |
| Run 4 | 89.06 | 0.31 | 90.62 | 0.22 |
| Run 5 | 90.62 | 0.26 | 84.37 | 0.51 |
| Average | 82.49 | 0.37 | 83.74 | 0.34 |

## Appendix B

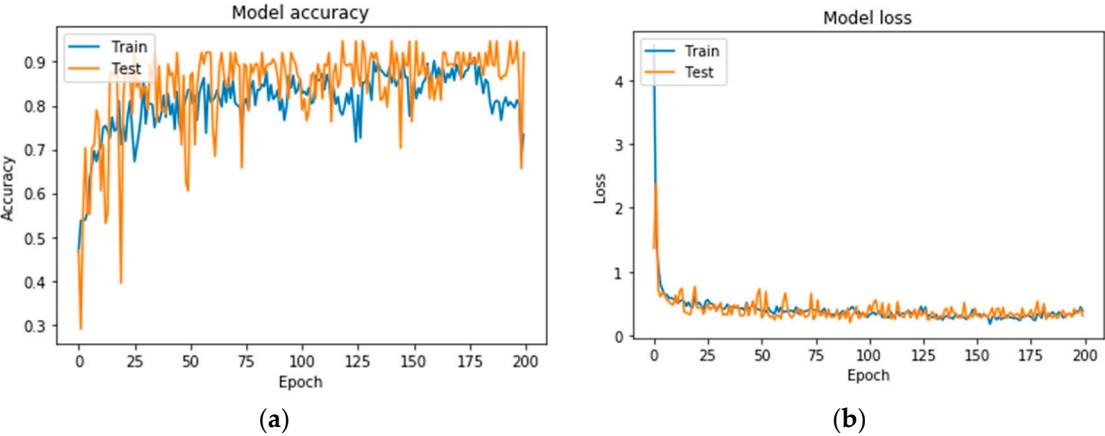

**Figure A1.** Precision curves for VGG16: (**a**) accuracy and (**b**) loss.

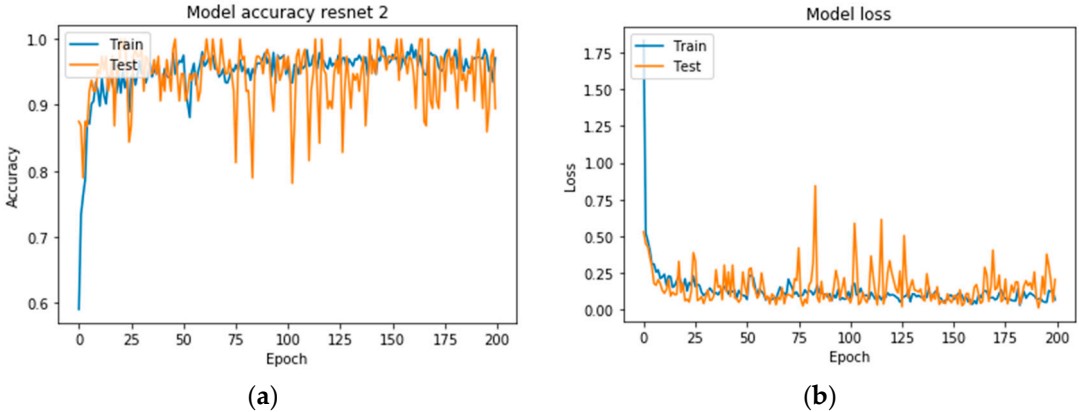

**Figure A2.** Precision curves for Resnet50 (**a**) accuracy and (**b**) loss.

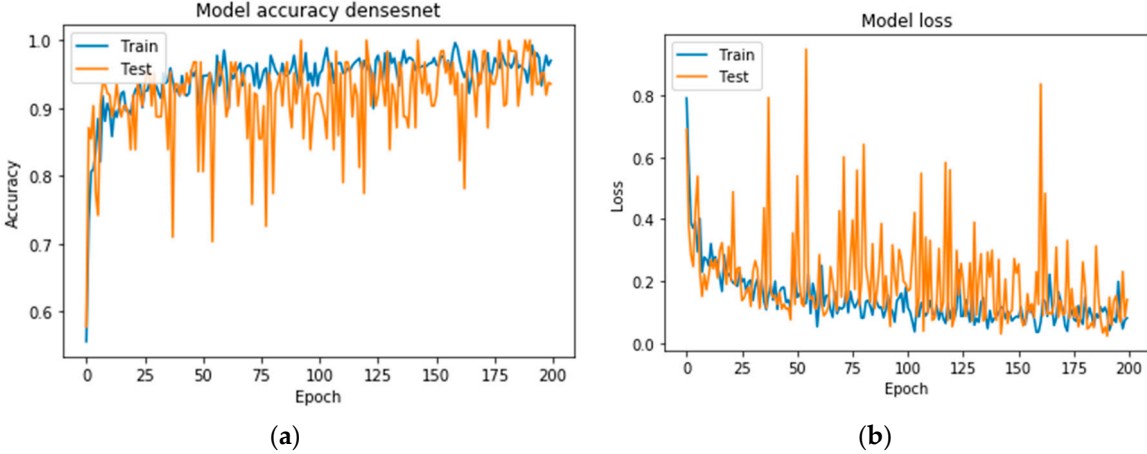

**Figure A3.** Precision curves for DenseNet (**a**) accuracy and (**b**) loss.

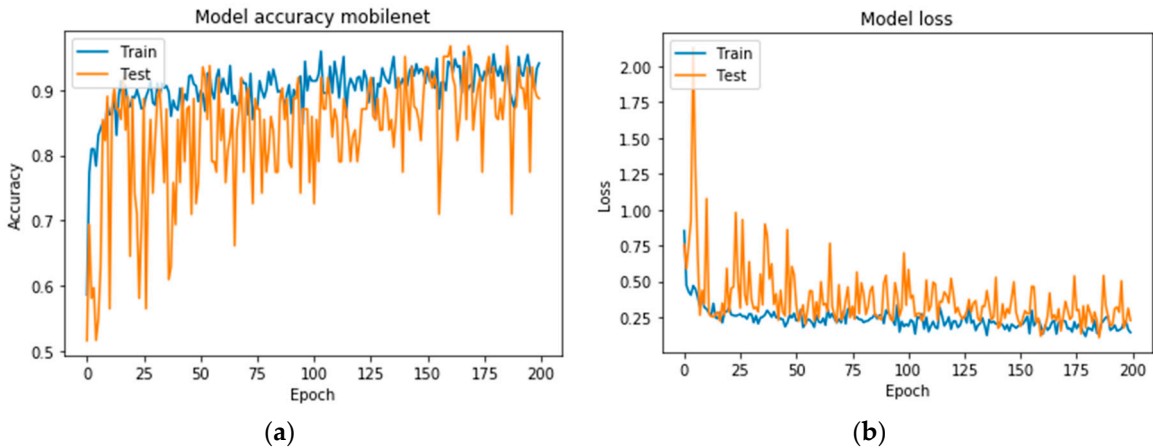

**Figure A4.** Precision curves for MobileNet (**a**) accuracy and (**b**) loss.

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
