# Peer review of "A Deep-Learning Approach for Diagnosis of Metastatic Breast Cancer in Bones from Whole-Body Scans"

_applsci, doi:10.3390/app10030997_

Round 1

Reviewer 1 Report

The article lacks novelty; it uses CNN to classify images of whole-body scans in patients who have breast cancer. The results are compared with trivial transfer learning models such as ResNet50, VGG16, MobileNet, and DenseNet The authors need to propose new mathematical formulations of the proposed approach and support with detailed experimentation results. The authors need to provide a link to the Google co-lab version of the implementation to appreciate the results. Reviewers need to be provided access to the dataset and results obtained by the approach, to validate the findings obtained.

Author Response

We sincerely thank the reviewers for their valuable and insightful comments which were of great help in revising the manuscript. Accordingly, the revised manuscript has been systematically improved with new implementations using Google Co-lab (as suggested by the first Reviewer) and additional information for novelty, evaluation of the results etc. Our responses (AR) to the reviewer’s comments (RC) based on the Review Report (Round 1) are clarified below: 

Also, English language and grammar editing have been accomplished by the authors and also by English native spoken editors to ensure the quality of the text.

RC1: The article lacks novelty; it uses CNN to classify images of whole-body scans in patients who have breast cancer. The results are compared with trivial transfer learning models such as ResNet50, VGG16, MobileNet, and DenseNet. The authors need to propose new mathematical formulations of the proposed approach and support with detailed experimentation results. 

AR 1: In the revised version of the manuscript, we amend the introduction section with an adequate description concerning the challenge and the aim of this research work, highlighting the innovation points according to the relevant literature.

The challenge focuses on building an intelligent and robust algorithm that automatically identifies whether a patient is suffering from metastatic breast cancer in bones or not, by looking solely at whole body scintigraphy images. The algorithm has to be extremely accurate because lives of people are at stake.

The main objective of this study is to create and demonstrate a customized, high-performance CNN-based classification tool for the identification of metastatic breast cancer in bones from whole body scans observation.

The novel contribution of this research work lies in the creation of a robust CNN model (being the most efficient deep learning method in medical image analysis) for bone metastasis diagnosis from breast cancer patients using whole-body scans. Following the literature review and the state-of-the-art deep learning methodology (as presented in sections 1.1 and 1.2), it is obvious that there is no previous work on metastatic breast cancer identification in bones from whole body scans using deep learning and CNNs.

Due to this fact, in this first research study in bone scintigraphy analysis for metastatic breast cancer, our leading intention was not just to propose a new mathematical formulation for this medical image analysis task, but to explore for the first time the capabilities of CNNs, being advanced deep learning methods with efficient capabilities in medical image analysis, (see review papers [47-50]) in classification of metastatic breast cancer in bones in the domain of nuclear medical imaging.

A Table with the cutting-edge breast cancer diagnosis using CNNs was added in section 1.1. Regarding all previous works, they were mainly devoted to primary breast cancer detection and segmentation from histopathological images and mammograms, whereas, only two research studies were previously conducted in bone metastasis classification from breast cancer patients in bone scintigraphy. The classification of metastatic breast cancer in bones in nuclear medicine imaging has been previously accomplished with the well-known multilayer neural networks, so no any previous work exists with CNN application. This is explicitly written in section 1.2.

In the revised manuscript, it is clearly reported that the novelty of this work is the development of a high-performance and robust CNN architecture for whole body scans classification in breast cancer bone metastasis.

The innovative and contributing points of this research study are modified and presented in bullets at the end of section 1.

The main reason for using the most popular CNN models for comparison purposes in nuclear medical imaging (ResNet, VGG, MobileNet and Densenet) is the fact that they have already been involved in medical image classification with significant contribution on the reported results (see [22-24,31,73-75]).

RC2: The authors need to provide a link to the Google co-lab version of the implementation to appreciate the results.

AR2: We are really grateful to the Reviewer for this insightful and useful comment which significantly helped us to improve our paper by implementing the proposed CNN architecture in a cloud-based fast computing environment.

In the revised manuscript, we have revised the Tables with the results produced by the implementation of our CNN in the Google co-lab. The new results are similar to the previous results accomplished by our PC in terms of classification accuracy and metrics.

Using Google co-lab (as a free cloud service with GPU support), the computation time of runs has been significantly reduced. The Figure 8 with the average computation times of CNNs was modified accordingly.

Following the reviewer’s suggestion, we provide all the results in doc files produced in google co-lab environment as well as a Google sheet with the validation results of the respective experiments. (please see link: https://docs.google.com/spreadsheets/d/1j-MaZvyhPrtYy7avrw2yPXAjZxoURKW_HUrlWtWy96M/edit?usp=sharing)

Also, in the folder “Google Colab Results”, a number of sub-folders is included regarding different settings of the CNN configuration (ie. batch size, number of epochs, dropout, pixel size). Please use the link https://drive.google.com/open?id=11hBkFpPPzBulwrwnD12KnCo7N9PAqIX4

These can be quite helpful for the Reviewer to have a thorough view and understanding on the results and their validation.

From the given Google sheet, it can be observed that 200 is the proper number of epochs for RGB mode to provide accurate results, as regards the best CNN (proposed) architecture.

Moreover, we provide access to the results (that were presented in the initially submitted paper) produced by a Home PC system, in the following link. (https://docs.google.com/spreadsheets/d/1dXvDq6nxuYtJ6D6cd9wiNqbvB4oQ4RMqKAdUJ6qvFoE/edit#gid=452327729).

Comparing the results produced by the Google co-lab environment with those performed by our PC, it is observed that the results are very similar for all the CNN architectures and the hyperparameter selection. The major difference lies in the run time, as it has been significantly reduced when the experiments were conducted in Google co-lab, following the Reviewer’s suggestion.

RC3: Reviewers need to be provided access to the dataset and results obtained by the approach, to validate the findings obtained.

AR3: Data availability: The dataset from Diagnostic Medical Center “Diagnostiki A.E.” was used under license for the current study, and is not publicly available.

However, an example dataset (80 images) of whole-body scans for metastatic breast cancer in bones is exceptionally provided in the following link (google drive) following the Reviewer’s request. Please use the link https://drive.google.com/open?id=11hBkFpPPzBulwrwnD12KnCo7N9PAqIX4

Code availability: The release of the code used for training and evaluation of the deep learning model is not feasible at this moment due to the research works that have been accomplished and are under submission. However, all experiments and implementation details are sufficiently described in the Methods section to allow independent replication with non-proprietary libraries. Several major components of our work are available in open source repositories including Keras (link) in Tensorflow (https://www.tensorflow.org, version 2.0). Data analysis was conducted in Python using the numpy (version v1.16.4), scipy (version 1.2.1), and scikit-learn (version 0.20.4) packages.

Our intention is the code becomes available in the research community (Github) just after the publications of the journal papers which are currently under submission, as regards bone scintigraphy analysis using CNNs.

In the revised version of the paper, we have included the results from the implementation of our CNN using the google co-lab environment, thus making it easy to validate the findings obtained.

Reviewer 2 Report

The authors presented a robust deep learning methodology for prediction purposes. The results seem promising with considerable impact. However, the clinical impact should be described more clearly as well as the application to the medical practice. 
Although comparison with CNN benchmarks is provided, it would be interesting to provide a table with recent research works in the literature on whole body bone metastasis diagnosis of BC patients. Hence, the novelty of the proposed framework will be clear. 

Some more details for the evaluation process of the classification procedure are missing. Which approach was used and why?

Author Response

Response: We thank you very much for your appreciation and positive feedback. Your valuable comments were of great help in revising the manuscript. Accordingly, the revised manuscript has been systematically improved with new implementations such as the use of Google co-lab (as suggested by the first Reviewer) and additional information for novelty, evaluation of the results etc. Our responses (AR) to the reviewer’s comments (RC), based on the Review Report (Round 1), are clarified below: 

Also, in the revised version of the manuscript, we performed English language editing and all minor grammar and syntax errors were carefully tackled.

RC1: The authors presented a robust deep learning methodology for prediction purposes. The results seem promising with considerable impact. However, the clinical impact should be described more clearly as well as the application to the medical practice.

AR1: We sincerely thank you for your positive and valuable feedback. In the revised manuscript, we revised the discussion of results section with a description of clinical impact and the application in the medical practice.

The following has been added at the end of the discussion of results section: “This is the first research study concerning the application of CNNs efficiently in bone metastasis diagnosis from breast cancer patients in bone scintigraphy. Following the proposed methodology, a robust CNN algorithm is created for bone metastasis diagnosis using whole-body scans from breast cancer patients. This algorithm can be implemented in any system without specific requirements, providing the nuclear medicine physician with an efficient and robust decision support tool to automatically identify whether a patient is suffering from breast cancer or not, by looking at whole-body scintigraphy images. Due to its high classification accuracy and performance, this algorithm has a significant positive impact in clinical diagnosis and decision making in nuclear medicine by providing diagnosis to metastatic breast cancer patients.

The proposed classification algorithm can be applied in any type of whole-body scans in bone scintigraphy and can be integrated into a CAD system of whole-body diagnosis or a clinical decision support system in nuclear medical imaging. Hence, it can be used by physicians in medical practice. “

The following has been added at the end of the conclusion section: “In future work, the authors intend to show how the proposed algorithm could be integrated into a CAD system of whole-body diagnosis in nuclear medicine and also provide evidence that the clinical decision support system could be generalized across continents.”

RC2: Although comparison with CNN benchmarks is provided, it would be interesting to provide a table with recent research works in the literature on whole body bone metastasis diagnosis of BC patients. Hence, the novelty of the proposed framework will be clear.

AR2: It is evident that much research has already been done for the detection and diagnosis of primary breast cancer in the past few years (see Review papers-). Some of the related papers are briefly discussed in the literature review.

Following the literature review for bone metastasis diagnosis from breast cancer in nuclear medicine, section 1.1 has been modified accordingly to separate the literature of deep learning in breast cancer diagnosis from that of deep learning in metastatic breast cancer in bones using scintigraphy images.

In the revised manuscript, the state-of-the-art breast cancer diagnosis is split into two subsections, the first one is devoted to the “Related Research in breast cancer diagnosis using CNNs” and the second one to the “Literature Review in bone metastasis diagnosis from breast cancer patients in nuclear medicine.”

A Table with the cutting-edge breast cancer diagnosis using CNNs was added in section 1.1. Regarding all previous works, they were mainly devoted to primary breast cancer detection and segmentation from histopathological images and mammograms, whereas, only two research studies were previously conducted in bone metastasis classification from breast cancer patients in bone scintigraphy using neural networks (no use of CNNs or other advanced deep learning structures was observed). This is explicitly written in section 1.1.

Also, subsection 1.3 was updated with a Table which summarizes the most recent applications of CNNs in bone metastasis recognition in nuclear medicine.

RC3: Some more details for the evaluation process of the classification procedure are missing. Which approach was used and why?

AR3: According to the relevant literature, concerning the evaluation of the classification performance, the most used approaches for evaluation purposes are: (i) the k-fold cross validation and (ii) the approach that splits the dataset to training, validation and testing data []. In the second method, the testing dataset is used for the evaluation/performance of the proposed classification method.

In our research work, following the second approach, a five stages methodology was proposed in which the dataset was split into training, validation and testing data. The training and validation phases were used for training the CNN and further investigating the proper parameters and structure of the CNN, which will be finally employed for the testing/evaluation process.

The dataset used for testing is an independent dataset and deployed for evaluation purposes to assess the predictive ability of the current model.

As far as the evaluation process is concerned, the performance of the model is examined considering the testing data and while this process uses some well-known evaluation metrics. The most used performance metrics in classification tasks are: precision, recall, sensitivity, specificity, f1-score and overall classifier accuracy (see [86,87,88,89]).

Concerning the evaluation of the classification process, a concise justification has been added in methodology section. (see Stage 5).

The following references have been added in the revised version.

Bishop, C. M., Hart, P. E. & Stork, D. G. Pattern recognition and machine learning. (Springer, c2006). Theodoridis, S., Koutroumbas, K. & Stork, D. G. Pattern recognition. (Academic Press, c2009). Labatut, V., Cherifi, H. Accuracy measures for the comparison of classifiers. Proceedings of the 5th International Conference on Information Technology (2011). Moustakidis et al. (2019) Application of machine intelligence for osteoarthritis classification: a classical implementation and a quantum perspective, Quantum Machine Intelligence, https://doi.org/10.1007/s42484-019-00008-3

In the revised version of the paper, we have included the results from the implementation of our CNN using the google co-lab environment, thus making it easy to validate the findings obtained.
